# Chromatin Accessibility and Transcriptional Differences in Human Stem Cell-Derived Early-Stage Retinal Organoids

**DOI:** 10.3390/cells11213412

**Published:** 2022-10-28

**Authors:** Melissa K. Jones, Devansh Agarwal, Kevin W. Mazo, Manan Chopra, Shawna L. Jurlina, Nicholas Dash, Qianlan Xu, Anna R. Ogata, Melissa Chow, Alex D. Hill, Netra K. Kambli, Guorong Xu, Roman Sasik, Amanda Birmingham, Kathleen M. Fisch, Robert N. Weinreb, Ray A. Enke, Dorota Skowronska-Krawczyk, Karl J. Wahlin

**Affiliations:** 1Viterbi Family Department of Ophthalmology, Shiley Eye Institute, University of California San Diego, La Jolla, CA 92093, USA; 2Department of Bioengineering, University of California San Diego, La Jolla, CA 92093, USA; 3Center for Translational Vision Research, University of California Irvine, Irvine, CA 92617, USA; 4Department of Biotechnology, California State University Channel Islands, Camarillo, CA 93012, USA; 5Center for Computational Biology and Bioinformatics, University of California San Diego, La Jolla, CA 92093, USA; 6Department of Obstetrics, Gynecology & Reproductive Sciences, University of California San Diego, La Jolla, CA 92037, USA; 7Department of Biology, James Madison University, Harrisonburg, VA 22807, USA

**Keywords:** CRISPR, organoids, retina, chromatin, transcription, gene-editing, ATAC-seq, RNA-seq, SIX6, VSX2

## Abstract

Retinogenesis involves the specification of retinal cell types during early vertebrate development. While model organisms have been critical for determining the role of dynamic chromatin and cell-type specific transcriptional networks during this process, an enhanced understanding of the developing human retina has been more elusive due to the requirement for human fetal tissue. Pluripotent stem cell (PSC) derived retinal organoids offer an experimentally accessible solution for investigating the developing human retina. To investigate cellular and molecular changes in developing early retinal organoids, we developed SIX6-GFP and VSX2-tdTomato (or VSX2-h2b-mRuby3) dual fluorescent reporters. When differentiated as 3D organoids these expressed GFP at day 15 and tdTomato (or mRuby3) at day 25, respectively. This enabled us to explore transcriptional and chromatin related changes using RNA-seq and ATAC-seq from pluripotency through early retina specification. Pathway analysis of developing organoids revealed a stepwise loss of pluripotency, while optic vesicle and retina pathways became progressively more prevalent. Correlating gene transcription with chromatin accessibility in early eye field development showed that retinal cells underwent a clear change in chromatin landscape, as well as gene expression profiles. While each dataset alone provided valuable information, considering both in parallel provided an informative glimpse into the molecular nature eye development.

## 1. Introduction

Retinogenesis represents the wave of development involving cell division, cell fate specification and differentiation of the neural retina into a complex multilayered tissue. While the mechanisms that control this process are not completely understood, they appear to be determined by spatiotemporally regulated gene expression and regulatory events. Advancements in the use of human pluripotent stem cells (PSCs) have created new avenues to study this process in humans. An intrinsic property of hPSCs is their potential to form most cell types in the body. Moreover, protocols have been developed for the directed differentiation of numerous cell and tissue types [1]. Although some differences remain, self-organizing organoids recreate structures with similar cellular composition, morphology and architecture as seen in vivo [2]. When grown in 3D, retinal organoids mimic in vivo visual system development and give rise to laminar organized retinas with all major classes of retinal cells [3,4,5].

A significant advantage to studying 3D retinal organoids lies in their experimental accessibility and the ease to which they can be visualized throughout development. Current limitations in the organoid field include correctly identifying the stage of development, accurately classifying cell types within the 3D system and determining how and if maturation of cells is achieved. Fluorescent reporters integrated at endogenous genes can address this by allowing for real-time observations of cellular and molecular changes throughout development [6,7,8]. For studying the mechanisms of eye development, coupling fluorescence to earlier stages of development (e.g., *PAX6*, *SIX6*) allows for confirmation and validation of retinal cells at early stages [9,10]. Further, *CRX* and *Nrl* reporters can identify the birth and migration of human and mouse photoreceptors, respectively [6,11,12]. Since *CRX* is expressed early in photoreceptor development it would allow for detection of early changes in chromatin accessibility and gene expression that could provide useful clues about cone and rod specification if coupled to single cell trajectory analysis.

Although transcriptional changes during retinogenesis have been studied in model organisms [6,13,14,15] and in humans [16,17,18], they have not encompassed all stages of human retinal development. Studies of PSC-derived retinal organoid gene expression suggest that organogenesis in vitro closely parallels in vivo development [7,10,19]. While transcriptomics studies have been performed on more mature retinal organoids, studies of early differentiation during early eye field and retinal progenitor stages have not been fully explored. In addition to gene expression, chromatin accessibility is a supplementary layer of complexity in gene regulation. The Assay for Transposase-Accessible Chromatin sequencing (ATAC-seq) method has identified temporal- and cell-specific changes in chromatin accessibility and regulatory elements in retinal cells in numerous species [20,21,22,23].

The aim of the current study was to elucidate the molecular changes leading up to retinal formation. To achieve this, a dual nuclear *SIX6*-p2A-h2b-eGFP (SIX6-GFP; early eyefield) and cytoplasmic *VSX2*-p2A-tdTomato (VSX2-tdTomato; retinal lineage) fluorescent reporter system was developed which enhanced our ability to track the transition periods for optic vesicles and retinas in real-time. An additional dual reporter was generated that incorporated a *VSX2*-p2A-h2b-mRuby3 (VSX2-h2b-mRuby3) nuclear reporter. When differentiated as retinal organoids, GFP was detected by day 15 and tdTomato (or mRuby3) as early as day 25. To correlate changes in chromatin accessibility and gene expression during early retinal development, ATAC-seq and RNA-seq, respectively, were performed on hPSCs (day 00) and days 5 to 25 organoids. Bioinformatic analyses identified global chromatin accessibility and gene expression differences between hPSCs and differentiated organoids. The chromatin landscape shifted from open to closed state for pluripotency associated genes (e.g., *NANOG*, *POU5F1* and *SOX2*) and vice versa for retina associated genes (e.g., *HES5*, *PAX6* and *SIX3*). Molecular changes in older optic vesicles were also identified by ATAC-seq of individual day 45 organoids and RNA-seq of day 35 and 45 retinal organoids. Decreased pluripotency and increased retinal progenitor gene expression confirmed a retinal fate, while additional pathway analysis identified “visual perception” as the main hallmark of day 45 organoids. Lastly, motif analysis revealed clear associations between accessible chromatin and developmentally relevant transcription factors. The current study correlates transcriptional landscape and chromatin structure with early eye formation, thereby providing new insight into human eye development.

## 2. Materials and Methods

### 2.1. Culture of Pluripotent Stem Cells

Commercially available IMR90.4 iPSCs (WiCell, Madison, WI, USA), further modified to integrate a high fidelity eSpCas9 and rtTA at the AAVS1 locus, were used with approval from the UC San Diego Institutional Review Board (IRB) committee. These were maintained in mTeSR1 media (Stemcell Technologies, Vancouver, BC, Canada) on 1% (*v*/*v*) growth-factor reduced Matrigel coated plates at 37 °C under hypoxic conditions (10% CO_2_/5% O_2_), as previously described [10]. To ensure cell quality, chromosomal integrity by copy number variation (CNV) analysis (Infinium HumanCore-24 v1.1 BeadChip, Illumina, San Diego, CA, USA) and PCR mycoplasma testing was performed [24].

### 2.2. Engineering Cell Reporters

The *VSX2*-p2A-tdTomato (VSX2-tdTomato) and *VSX2*-p2A-h2b-mRuby3 (VSX2-h2b-mRuby3) donors for CRISPR gene editing was created by PCR amplifying a 1157 bp fragment of the *VSX2* locus (F1: CGATGTACGGGCCAGATATAGCATGTGCTGTGGCCTGGAA, R1: GCGCAAAGCTTGGTACCAAATGGGCCTGCACAGTGTCTTCA from human genomic DNA (Phusion Flash Polymerase #F-548; Thermo Scientific, Waltham, USA) and inserting that into a custom donor vector by Gibson Assembly (#E5510; NEB, Ipswich, MA, USA) [25]. A p2A (porcine teschovirus-1 2A) polycistronic sequence coupled to tdTomato (or h2b-mRuby3) was then amplified and inserted before the stop codon by Gibson assembly followed by overnight Dpn1 digestion. The resulting donor cassette had a 591 base-pair (bp) left homology arm, a p2A coupled to tdTomato (or h2b-mRuby3) and a 566 bp right homology arm flanking the stop codon. A guide RNA (gRNA) directed at the stop codon (GGA GGA CAT GGC TTA GGT CA) was inserted by site-directed mutagenesis into the same plasmid between a human U6 promoter and a gRNA scaffold. After transformation into Stable *E. coli* (#C3040; NEB, Ipswich, MA, USA) cells, colonies were miniprepped, sequence verified and plasmid DNA was prepared for transfection using a [10]). SIX6-p2A-h2b-eGFP (SIX6-GFP) reporter PSCs were made as previously described [10]. These were subsequently transfected into undifferentiated PSCs with the VSX2 donor vectors using the Neon transfection system (Thermo Scientific, Waltham, MA, USA) as previously described. To induce expression of eSpCas9, IMR90.4 hPSCs were treated with doxycycline (1μg/mL) overnight prior to transfection. For genotyping, clonal PSC colonies were picked, expanded, replica plated and lysed in Quick Extract buffer (#QE09050; Lucigen, Middleton, USA) followed by PCR using one oligonucleotide flanking the left homology arm (VSX2_PrBGT_F1: GCC CTC TGG GCC TAT ATC TGA GAA CAG CAC) and one oligonucleotide nested within the p2A insert (P2A_GTrev2: AGG CTG AAG TTA GTA GCT CCG CTC CC). Oligos flanking both homology arms (VSX2_het828F: AGG AAC TGA GGG AGA ACA GCA TT, VSX2_het828rev: AGG CTG AGA CTT CCC AAC TGT AG) were used to confirm homozygosity. PCR products were verified by Sanger sequencing (Figure 1A and Appendix A).

### 2.3. Generation of Retinal Organoids

Retinal organoids were differentiated as previously described [10] Briefly, cells were passaged with Accutase (#A6964; Sigma-Aldrich, St. Louis, MI, USA) for 12 min at 37 °C, dissociated to a single cell suspension, quenched with mTeSR1 plus 5 μM blebbistatin (#B0560; Sigma-Aldrich, St. Louis, MI, USA), pelleted at 80× *g* for 5 min at room temperature, resuspended in mTeSR1 plus blebbistatin and 1000–3000 cells plated in 50 μL’s per well in 96-well round bottom plates (#650180; Greiner, Frickenhausen, Germany) (day 00). For days 1 to 3, organoids were fed with 50μL’s of BE6.2 (B27 plus E6 at 2x) plus a final concentration of 1% (*v*/*v*) Matrigel-GFR and 3 µM IWR-1e (#681669; EMD Millipore, Burlington, NJ, USA). For days 4 and 5, 50% of the media was exchanged with fresh BE6.2 plus Matrigel and IWR-1e. At day 6, only BE6.2 and Matrigel were exchanged. Cells were maintained for 6 days in hypoxia (10% CO_2_/5% O_2_) before being transferred to normoxia (5% CO_2_/20% O_2_). At days 8 and 10, BE6.2 media plus 300 nM smoothened agonist (SAG; #566660; EMD Millipore, Burlington, NJ, USA) was exchanged. To separate the optic vesicles, organoids were cut with electrolytically sharpened tungsten needles at days 10 through 16. Dissected organoids were maintained on non-tissue culture treated petri dishes fed with long-term retina (LTR) media plus SAG until day 17, then without SAG and with 500 nM retinoic acid until day 45. DAPT (10 µM) was added starting at day 29 until day 42. For long term maintenance, organoids were fed with LTR plus RA from day 20 onward.

### 2.4. Flow Cytometry

Day 45 SIX6-GFP/VSX2-tdTomato organoids were screened for GFP and tdTomato fluorescence using an ImageXpress Micro Confocal Imaging System (Molecular Devices, San Jose, CA, USA). Organoids were pooled (n = 10–20) and dissociated into single cell suspensions. Organoids were incubated in Accutase for 5 min, swirled gently and incubated for another 5 min. Organoids were washed 3 times in PBS (Ca^2+^/Mg^2+^ free), gently dissociated in sort buffer (Ca^2+^/Mg^2+^ free PBS, 1mM EDTA, 25mM HEPES pH 7.0, 1% FBS) and filtered (40 µm pore) for a final concentration of 5 × 10^6^ cells/mL. Cells were sorted at the Flow Cytometry Core Facility at the La Jolla Institute for Immunology on a FACSAria-3 (BD Biosciences, Franklin Lakes, NJ, USA).

### 2.5. Sample Collection

To screen for fluorescence, organoids were imaged in 96-well round bottom plates (#650180; Greiner, Frickenhausen, Germany) using an ImageXpress under widefield settings. For RNA-seq, 2 × 10^5^ SIX6-GFP/VSX2-tdTomato undifferentiated control hPSCs (n = 3) and differentiated organoids were collected based on morphology (days 05 to 45), GFP single fluorescence (days 15 to 20) and GFP-tdTomato dual fluorescence (days 25, 35 and 45). Organoid samples were pooled as triplicates (n = 8–20 organoids/replicate days 05 to 25, n = 2–5 organoids/replicate days 35 and 45, washed three times with PBS (Ca^2+^/Mg^2+^ free) and frozen at −80 °C. For ATAC-seq, triplicate samples of SIX6-GFP/VSX2-tdTomato hPSCs (2 × 10^5^ cells; day 00), SIX6-GFP/VSX2-tdTomato (n = 8–20 pooled organoids/replicates for days 5 through 25 and n = 1 organoid/replicate for day 45) were collected in Synth-a-Freeze Cryopreservation Media (#A1254201; Thermo Scientific, Waltham, MA, USA) and stored in liquid nitrogen until preparation (see nuclei isolation below).

### 2.6. Preparation and Imaging of Sectioned Organoids

For frozen cryosections, organoids were collected, washed in PBS (Ca^2+^/Mg^2+^ free) three times, fixed in 4% paraformaldehyde in 0.1M Sorenson’s Buffer for 1 h at room temperature, preserved in increasing concentrations of cold sucrose (5%, 6.75% and 12.50%) in 0.1 M Sorenson’s Buffer for 30 min each and in 25% sucrose overnight at 4 °C. Organoids were embedded in cryomolds (#25608-922; Tissue-Tek, Torrance, CA, USA) using Optimum Cutting Tissue Temperature (OCT) compound (#4583; Tissue-Tek, Torrance, CA, USA), sectioned on a Leica CM1860 cryostat microtome at 10 μm thickness, mounted onto Superfrost Plus Micro Slides (#48311-703; VWR, Radnor, PA, USA) and stored at −80 °C. To visualize endogenous GFP and tdTomato fluorescence, imaging was performed on an ImageXpress microscope.

### 2.7. qRT-PCR

For quantitative PCR (qPCR), cDNA was prepared from day 45 organoid RNA using the Bio-Rad iScript Reverse Transcription Super Mix for RT-qPCR (#1709941; Bio-Rad, Hercules, USA). qPCR was performed on a Bio-Rad CFX Connect Real-Time PCR Detection System (Bio-Rad, Hercules, CA, USA) using Bio-Rad SYBR Green Supermix (Bio-Rad, Hercules, CA, USA). Biological replicates (n = 3) were analyzed in technical duplicates and normalized to the expression of the reference gene hRPL27 and relative expression was determined by the ΔΔCt method. The following primers were used: hRPL27_123_F: ATC GCC AAG AGA TCA AAG ATA A, hRPL27_ 123_rev: TCT GAA GAC ATC CTT ATT GAC G, SIX6_198_F: ACC CCT ACG CAG GTG GGC AA, SIX6_198_rev: TGA AGT GGC CGC CTT GCT GG, VSX2_122_F: GGC GAC ACA GGA CAA TCT TTA, VSX2_122_rev: TTT CCG GCA GCT CCG TTT TC.

### 2.8. Nuclei Isolation and ATAC-seq Libraries

ATAC-seq was performed as previously described, with modifications [26]. Lysis was performed in fresh nuclei permeabilization buffer (5% BSA, 0.2% IGEPAL-CA630, 1 mM DTT, 1x complete EDTA-free protease inhibitor in PBS (Ca^2+^/Mg^2+^ free) on ice, filtered through a 40 μM nylon mesh and centrifuged for 5 min at 500× *g* at 4 °C. Nuclei were resuspended in cold tagmentation buffer (33 mM Tris-acetate, 66 mM potassium acetate, 11 mM magnesium acetate and 16% DMF) and counted. A total of 20,000–50,000 nuclei were used for tagmentation by adding Tagment DNA Enzyme 1 (Illumina, San Diego, CA, USA), incubated for 30 min with 500 rpm at 37 °C. Fragments were purified using the QIAquick PCR Purification Kit (Qiagen, Hilden, Germany) and amplified by PCR. Size selection was performed using AMPure XP beads (Beckman Coulter, Brea, CA, USA) to remove short DNA fragments. Libraries were quantity and quality checked using a 4200 TapeStation (Agilent Technologies, Santa Clara, CA, USA). Sequencing was performed on a HiSeq4000 (50-bp paired end, Illumina, San Diego, CA, USA) at the Institute for Genomic Medicine at the University of California San Diego (UCSD IGM).

### 2.9. RNA Isolation and RNA-seq Libraries

Total RNA was isolated using the RNeasy Plus Mini Kit (Qiagen, Hilden, Germany) per manufacturer’s instructions. RNA quantity and quality were analyzed with a NanoDrop 2000 (Thermo Scientific, Waltham, MA, USA) and 4200 TapeStation (Agilent Technologies, Santa Clara, CA, USA), respectively. RNA-seq libraries were generated for days 00 to 25 using the NEBNext Ultra RNA Library Prep Kit for Illumina (#E7530, NEB, Ipswich, MA, USA) and days 35 to 45 using the TruSeq RNA Library Prep Kit v2 (RS-122, Illumina, San Diego, CA, USA). Sequencing was performed on a HiSeq4000 (150 paired-end, Illumina, San Diego, CA, USA) at Genewiz (South Plainfield, NJ, USA) for samples day 00 through 25 and at the UCSD IGM for days 35 and 45.

### 2.10. ATAC-seq and RNA-seq Data Analyses

RNA-sequencing was performed by Genewiz (days 0 to 25) to obtain 40 million reads and the IGM at UCSD (for days 35 and 45) to obtain 35 million reads in biological triplicate. Analyses were performed by the Center for Computational Biology and Bioinformatics (CCBB) at UCSD. Quality control of the raw fastq files was performed using the software tool FastQC v0.11.3. Sequencing reads were trimmed with Trimmomatic v0.36 [27] and aligned to the human genome (GRCh37.p13) using STAR aligner v2.5.3a. Read quantification was performed with RSEM v1.3.0 [28] and the Gencode release 19 annotation [29]. The R BioConductor packages edgeR [30] and limma [31] were used to implement the limma-voom method [32] for differential expression analysis. In brief, lowly expressed genes—those with counts per million (cpm) < 1 in at least 3 of the samples were filtered out and then trimmed mean of M-values (TMM) normalization [33] was applied. The experimental design was modeled upon day (~0 + day). The voom method was employed to model the mean-variance relationship in the log-cpm values, after which lmFit was used to fit per-gene linear models and empirical Bayes moderation was applied with the eBayes function. Significance was defined by using an adjusted *p*-value cut-off of 0.05 after multiple testing correction [34] using a moderated t-statistic in limma.

For ATAC-seq, peak calling, peak height measurements and motif analyses were done using a sliding window of 400 bp in HOMER (Hypergeometric Optimization of Motif EnRichment) [35]. Normalization and differential analysis (DA) were done in R using limma [31]. Principal component, hierarchical cluster analyses and scatterplots were designed using BioVinci v1.0.0 (BioTuring, Inc., San Diego, CA, USA), ATAC-seq peaks were visualized with IGViewer v2.5.3 [36], statistical analyses were completed in Galaxy/Cistrome [37] and figures were generated in Prism (v5.01; GraphPad Software, San Diego, USA). Pathway analysis was performed using DAVID (v6.8) [38] and significance was accepted at FDR < 0.1 for analysis of DEG/DARs and an FDR < 0.001 for analysis of DEGs only. Data sets are compliant per MINSEQE standards and available at NCBI SRA (PRJNA771414; PRJNA876944).

## 3. Results

### 3.1. Generation of a Dual Fluorescent Reporter hPSC Line for Tracking Retinal Development

SIX6 is required for initiation of early eye development [39,40] while VSX2 is involved in the maintenance and proliferation of retinal progenitors and bipolar cell specification [41]. Previously we used SIX6-GFP and SIX6-GFP/POU4F2-tdTomato reporters to identify optic vesicles before and after retinal ganglion cell (RGC) formation [10]. In the current study, we introduced a VSX2-tdTomato reporter into the SIX6-GFP PSC reporter line (Figure 1A,B) to identify retinas at the earliest possible stage of ocular development. Following CRISPR-mediated integration of VSX2 with p2A-tdTomato (VSX2-tdTomato) or p2A-h2b-mRuby3 (VSX2-mRuby3), homozygous clones were verified by Sanger sequencing (Appendix A). The top genomic off-target sites for SIX6 and VSX2 guide RNAs (gRNAs) were sequence verified to rule out off-target genome editing (Appendix A). Chromosomal integrity was analyzed by copy number variation (CNV) analysis, which has a 5-fold greater resolution than standard G-band karyotyping, and no karyotypic changes were observed (Appendix A). To validate the utility of IMR90.4 SIX6-GFP/VSX2-tdTomato dual reporter PSCs we generated 3D retinal organoids using previously published methods (Figure 1C). During differentiation, stem cells and developing organoids were observed by morphology and fluorescence up to day 45 (Figure 1D–F). By day 05, organoids had a rounded morphology with a phase bright surface. Neural vesicles appeared between day 08 through 10, after which time they were manually excised for further growth and differentiation. SIX6-driven nuclear GFP fluorescence was detected in presumptive optic vesicles by day 15 (Figure 1D and Appendix A) and by day 27, labeling of SIX6-GFP and VSX2-tdTomato were present throughout the organoid with GFP expressed in nuclei and tdTomato expressed throughout the cytoplasm of radial oriented progenitors (Figure 1G and Appendix A). The location of VSX2+ cell nuclei was verified using a separate nuclear encoded VSX2-mRuby3 reporter iPSC line (Figure 1H,J,L). By day 45, VSX2-tdTomato fluorescence was detected over large areas of the organoid, indicating an expansion of proliferating cells (Figure 1I,J). Imaging of day 45 cryosections revealed that tdTomato+ organoids were laminar organized with a presumptive outer neuroblast (ONBL) layer (Figure 1J). Imaging of nuclear localized VSX2-mRuby3+ organoids similarly showed prominent signals in the ONBL with co-localization of SIX6+ and VSX2+ nuclei. Though VSX2 itself is normally expressed in cell nuclei, the VSX2 reporter harboring cytoplasmic fluorescence (Figure 1G,I,K) was helpful for visualizing the shape and alignment of progenitor cells. By day 120, SIX6-GFP and VSX2-tomato signals had unique expression patterns with SIX6 detected throughout the retina and VSX2 which became restricted to the outer aspect of the inner nuclear layer [42], where bipolar cell nuclei typically reside (Figure 1K,L). This was particularly evident in VSX2-mRuby3 organoids that had mRuby3 labelled cell nuclei (Figure 1L). The reproducibility among replicates (Appendix A) validated these dual fluorescent reporters for use in tracking retinal cells in real-time.

### 3.2. Chromatin Accessibility and Gene Expression Changes during the Stem Cell to Retinal Organoid Transition

To assess molecular changes during early ocular development, ATAC- and RNA-seq were performed on developing SIX6-GFP/VSX2-tdTomato+ organoids at 5-day intervals from days 00–25, including the onset of SIX6-GFP by day 15 and tdTomato fluorescence at day 25 (Figure 2). Principal component analysis (PCA) from 66,201 peaks from ATAC-seq at the various time-points showed temporal differences in chromatin accessibility with tight clustering among replicates (Figure 2A). The landscape of chromatin accessibility at the different time points (days 00–25) was evident between aligned genomic tracks (Figure 2B; arrows). ATAC-seq peaks within/near known genes indicated a high proportion of peaks at intergenic, intronic and promoter regions and less in 3′UTR, 5′UTR, exonic, non-coding and transcription termination site (TTS) regions (Figure 2C). A PCA plot of RNA-seq samples showed transcriptional differences between time points with tight clustering among replicates (Figure 2D). Hierarchical cluster analysis (HCA) of the top 100 most highly differentially expressed genes (DEGs) indicated transcriptional differences between stem cells (day 00) and optic vesicles (day 25) (Figure 2E; Appendix A). We then sought to correlate SIX6-GFP and VSX2-tdTomato fluorescence with changes in open chromatin and mRNA using ATAC-seq and RNA-seq. Chromatin accessibility at the transcription start site (TSS) of SIX6 and VSX2 showed a peak at the SIX6 TSS from day 05 through day 25; whereas no peak was detected at any time point at the VSX2 TSS (Figure 2F, left). SIX6 mRNA was upregulated by day 10 whereas VSX2 was only weakly expressed (Figure 2F, right). To investigate the transition of stem cells through early retinal differentiation, markers for stem cells and different stages of development were analyzed (Figure 2G,H). Some pluripotency markers, such as NANOG and POU5F1(OCT4), had open chromatin peaks and expression levels that both decreased over time. Conversely, chromatin accessibility at the SOX2 locus remained elevated and mRNA expression increased over time (day 25/day 00 log_2_ FC 1.98), consistent with its role in neural progenitors. Over time, the retinal progenitor markers HES5, LHX2, NOTCH and SIX3 had increased peaks near the TSS, whereas HES1 and PAX6 showed only subtle changes (Figure 2H, left). RAX and SUFU were largely unchanged; however, many of the retinal progenitor markers showed a general increased pattern of mRNA expression from day 00 to day 25 (Figure 2H, right). Collectively, these results indicated that changes in both gene expression and chromatin accessibility occur from pluripotency (day 00) through optic vesicle stages of development (day 25).

ATAC- and RNA-seq analyses involved stem cell to early eye field (day 15/day 00), stem cells to retinal progenitors (day 25/day 00) and early eye field to retinal progenitor (day 25/day 15) transitions. Differentially accessible regions (DARs) from ATAC-seq were mapped to their cognate genes, compared to differentially expressed genes (DEGs) from RNA-seq for days 00, 15 and 25 and represented as a Venn diagram to show areas of overlap or uniqueness (Figure 2I). This identified a total of 110 genes between day 15/day 00, 669 genes between day 25/day 00 and 308 common DEG/DARs that changed both mRNA levels and chromatin accessibility with respect to day 00. Genomic annotations of the DARs indicated a higher degree of change in intergenic (40.38% day 15/day 00, 51.85% day 25/day 00 and 40.81% common) and intronic regions (47.60% day 15/day 00, 37.04% day 25/day 00 and 51.02% common), a smaller proportion in promoter regions (8.65% day 15/day 00; 7.41 % day 25/day 00 and, 4.08% common) and low representation in the remaining regions (0–3.70% for 3′UTR, 5′UTR, exonic, non-coding and TTS) (Figure 2J). Several genes with significant correlation between DARs and DEGs were AXIN2, NOG and ESRP (Figure 2K,L). In the day 25/day 00 comparison, AXIN2, a key component of canonical Wnt Signaling, had two significant DARs, one in the intergenic region (right arrow) that decreased during differentiation and one in the third intron (left arrow) that increased. At the mRNA level, AXIN2 gradually decreased by day 25 (log_2_FC −1.11) consistent with its role in ocular development [43] and stem cells [44], thus the opposing peak patterns may suggest different regulatory uses during development. For the day 15/day 00 comparison, a significant peak at the transcription termination site (TTS) of NOG (noggin) was identified. Noggin aids in the induction of neural precursors [45] and is expressed in the vertebrate retina [46]; an increase in both the peak and mRNA expression for NOG suggests differentiation towards a neural lineage. For the common DARs and DEGs between day 15/day 00 and day 25/day 00, a DAR was identified at the transcription start site (TSS) of ESRP1 (Epithelial Splicing Regulatory Protein 1), which is involved in pluripotency [47]. There was a decrease in the peak at the promoter (Figure 2K) during differentiation and a sharp decrease in gene expression (Figure 2L) from day 05 to 15. Since RGCs are one of the first retinal neurons to be born, we compared ATAC-seq DARs and RNA-seq DEGs for RGC markers. Based on the ATAC-seq and RNA-seq depicted in Appendix A, ATOH7, NEUROG2, POU4F2 and SNCG showed low levels of DEG/DAR expression at the earlier time points (day 0–25) that increased in the later time points (day 35–45). These molecular changes indicate a likely transition state from pluripotency to neural retina.

### 3.3. Signaling Pathways during Optic Vesicle Differentiation

To identify pathways enriched during differentiation, DAVID analysis was performed using a gene list of DARs secondarily filtered with DEGs (Figure 2I, Appendix A). 110 genes, present in 677 KEGG pathways, GO terms and UP keywords were identified from the day 15/day 00 comparison, of which 9 pathways were significant (FDR < 0.1; Appendix A). On the other hand, 669 genes were identified from the day 25/day 00 comparison in 1487 pathways, 68 of which were significant. Next, we expanded our search to include all identified genes that were detected by RNA-seq. Moreover, a cross-comparison of DEG time points classified 288 genes in common between the day 15/day 00 and day 25/day 00 datasets, 236 genes unique to day 15/day 00 and 234 unique genes to day 25/day 00 (Appendix A). For more comprehensive pathway analysis, we used significant DEGs to identify pathways belonging to individual time points in Appendix A.

From the DEG/DAR day 00/day 15 comparison (Figure 3A), pathways from nervous system development, neuron projection and signal transduction were identified. Nervous system development was also present in the day 00/day 25 comparison (Figure 3B), along with regulation of axonogenesis and axon guidance, both indicative of maturing retinal progenitors. Pathway analysis was performed again with DEGs alone and from these, seven (axon guidance, Rap1 signaling, regulation of pluripotency, focal adhesion, cAMP signaling, MAPK signaling and calcium signaling) were found to be common in both time point comparisons. One of these was “signaling pathways regulating pluripotency of stem cells” (Figure 3C). Since many genes involved in “pluripotency” are also expressed later (e.g., SOX2) their presence may be indicative of a progenitor-like state, not necessarily pluripotency. Gene expression changes also showed a divergence from early time points (days 00) to later time points during differentiation (days 15–25). Fifty-six genes were differentially expressed at days 15 and 25 as compared to day 00 (Figure 3C, left). The DEGs from the pluripotency pathway were further categorized into TGFβ signaling, Wnt signaling, kinase signaling and transcription factors (Figure 3C). The “axon guidance signaling” pathway also showed a high similarity between genes at day 15/day 00 and day 25/day 00 comparisons (Figure 3D). Further subcategorization of “axon guidance signaling” into ephrin, netrin and semaphorin families of genes, however, showed progressive differences over time (Figure 3D). Thus, “axon guidance signaling” may be indicative of neurite outgrowth during retinal development. Pathway analysis of DARs (Appendix A) also identified axon guidance signaling as important. Collectively, these trends indicated a shift from pluripotency towards a neural fate.

### 3.4. Molecular Differences during Early Retinal Cell Specification

VSX2-tdTomato fluorescence was first detected at day 25 (Figure 1F); however, mRNA expression levels were low and open chromatin peaks near the TSS of the VSX2 gene were below the threshold for detection (Figure 2F). To further validate expression of VSX2, ATAC- and RNA-seq at later time points (days 35 and 45) were performed. We analyzed the datasets from days 00 to 25 and days 35 to 45 independently when determining gene expression to minimize batch effect. A PCA plot of transcripts from retinal organoids at days 35 and 45 showed a clean separation of the time points based on gene expression variation (Appendix A). Of the 15,063 genes expressed at both time points, 8.6% of the genes were significantly downregulated (1294 genes) and 9.9% of the genes were significantly upregulated (1496 genes) in day 45 as compared to day 35 (Appendix A). To observe inter-organoid differences, we performed ATAC-seq on individual SIX6-GFP/VSX2-tdTomato+ day 45 organoids. Global genomic peaks showed little variation between organoids (Appendix A) and high concordance between the replicates (Pearson correlation range of 0.80–0.97; Appendix A). To investigate gene driven GFP and tdTomato fluorescence, chromatin accessibility and gene expression of SIX6 and VSX2 were analyzed. Like analyses of the day 25 time point, a peak at the TSS for SIX6, but not VSX2, was detected (Figure 4A, left); however, RNA-seq analysis showed increased expression of both SIX6 and VSX2 at day 45 as compared to day 35 (Figure 4A, right). To quantify the number of SIX6-GFP and VSX2-tdTomato positive cells, flow cytometry was performed (Figure 4B). At day 15, organoids expressed GFP but not tdTomato, which was validated by flow cytometry (36.70 ± 1.30% GFP+ and 0.00% tdTomato cells; Figure 4B). Similarly, day 45 organoids had both GFP and tdTomato fluorescence which was detected by flow cytometry (10.55 ± 1.65% GFP+ and 24.55 ± 4.15% dual GFP/tdTomato+ cells; Figure 4B). All day 45 tdTomato+ cells co-expressed GFP, but not all GFP+ cells expressed tdTomato. Additionally, qRT-PCR of day 45 organoids detected expression of both SIX6 and VSX2 (Figure 4C), further verifying that fluorescence is dependent on SIX6 and VSX2 expression.

In early development, VSX2 is involved in the maintenance and proliferation of retinal progenitors and in later stages it participates in bipolar cell differentiation and blocks photoreceptor fate [41,48,49]. To determine if retinal specification via VSX2 expression progressed towards bipolar cells at these time points, bipolar expressed genes (CABP5, GRM6, KCNG6, LHX4 and VSX1) were analyzed. ATAC-seq analysis showed small or no TSS peak changes for any gene at day 45, however, increases in mRNA were detected from days 35–45 (Figure 4D). In mice, upstream regulatory elements for VSX2 have been identified and are required for the development of bipolar cells [50,51,52,53]. At day 45, a comparable conserved area in the human intergenic region upstream of VSX2 showed no peak at the mouse bipolar cell-specific cis-regulatory element (CRE) described by Kim et al. [51] (Figure 4E, arrowhead). While it is possible that this site does not exist in humans, a more likely explanation is that mature bipolar cells do not yet exist. Another regulatory unit for VSX2 that was also described in mice is the core transcriptional regulatory circuitry super enhancer (CRC-SE) [50] which appeared at day 45 (Figure 4E, arrow), but not at earlier time points (Appendix A, arrow). CRC-SEs can bind to their own loci and self-regulate [54,55] and the VSX2 consensus sequence TAATTA [50,56] was located within the peak (Appendix A, arrow). At day 45, expression from the surrounding genes was compared and VSX2 (645.45 ± 116.07 CPM) had a 4.53 log_2_ fold change higher expression than LIN52 (28.02 ± 3.72 CPM) (Figure 4F) whereas LIN52 had a range of 2.63 to 7.93 log_2_ fold change higher expression than VSX2 during days 00 to 25 (Appendix A). Additionally, a separate peak further upstream near the LIN52 gene was interrogated for the VSX2 consensus binding sequence and three potential sites were identified (Appendix A, *), suggesting this may be an additional CRC-SE VSX2 binding site. No other VSX2 consensus binding sequences were found in the intergenic region between LIN52 and VSX2. LIN52 is a component of the dimerization partner, RB-like, E2F and multi-vulval class B (DREAM) complex which coordinates gene expression during the cell cycle [57] and drosophila lin-52 mutants had eye defects [58]. The unique peak identified may be involved in VSX2 regulation and expression, yet additional analyses are necessary to substantiate this finding. Furthermore, day 45 organoids may have early expression of photoreceptor and bipolar cells but are mainly composed of retinal progenitor cells.

While the organoids used in the current study were still very young, photoreceptors were just beginning to form. To look for early signs of chromatin accessibility at photoreceptors genes, we analyzed chromatin peaks for several photoreceptor genes (AIPL1, ARR3, CNGB1, CRX, GNGT1, GUCA1A, GUCA1B, NRL, PDE6B, PDE6C, PDE6H, PROM1, RCVRN and RS1) and observed little to no TSS peaks, however, we did detect slightly elevated gene expression from days 35–45 (Figure 4G). Given that photoreceptor numbers were still limited at this stage, this is not surprising. Conversely, retinal progenitor genes (HES1, HES5, LHX2, NOTCH1, PAX6, RAX, SIX3 and SUFU) had peaks at their respective TSSs and high levels of expression (Figure 4H). From days 00–25, pluripotency marker expression declined (Figure 2G). NANOG and POU5F1 had no TSS peaks; whereas SOX2 still retained a peak at day 45 (Figure 4I, left). This corresponds with the gene expression profiles of NANOG and POU5F1 that were absent and SOX2 that was elevated (Figure 4I, right). Additionally, genes identified from a published fetal retina set (fetal day 59; GEO: GSE142526) were used for comparison to our organoid dataset [18]. Cell-type markers for progenitor, transition stage 1 (T1), amacrine, retinal ganglion and cone cells were expressed at both days 35 and 45 (Appendix A). Specifically, PRDM13 (amacrine/transition 2), ATOH7 (T1) and OTX2 (photoreceptor/transition 3 cells) increased and SOX2 (progenitor cells) decreased from days 35–45 (Appendix A). Pathway analysis (GO–Biological Processes) from significant DEGs between days 35–45 identified 10 significant pathways (FDR < 0.001) (Figure 4J; Appendix A). The pathway ‘visual perception’ (FDR = 4.55 × 10 ^−19^) at day 45 was further categorized into phototransduction, visual function, or structural gene pathways (Figure 4K). Gene expression analysis also indicated a transition from retinal progenitor to a more mature state for early born cell types. Overall, this study showed that SIX6 and VSX2 dual reporter retinal organoids can be effective for monitoring transitions in developing retinal organoids and for correlating changes in gene expression and chromatin accessibility.

### 3.5. Motif Discovery and Analysis

To characterize transcription factor and gene motifs within the ATAC-seq dataset, we utilized HOMER to perform peak calling and motif discovery on peak tag directories from each time point [35]. HOMER identified several hundred known enriched motifs, of which 20 were represented in a heat map signifying motif enrichment with respect to fold change from background sequences (Figure 5A and Appendix A). We observed that pluripotency-related motifs (such as and OCT4-SOX2-TCF-NANOG) were highly enriched at earlier time points (day 00 and day 05) whereas motifs for retinal TF genes (e.g., DLX1, DLX2, LHX2, PAX6, SOX4 and SOX9) were abundant at later time points, particularly at day 45. To investigate differential changes in peak heights between day 00/day 25 and day 00/day 45 volcano plots were generated using ATAC-peak height log_2_(fold change) and –log10(*p*-value). Day 00/day 45 showed an increase in chromatin accessibility at the TSSs of known retinal progenitor genes (e.g., NEUROG2, PAX6) and RGC genes (e.g., ISL1 and ATOH7). This coincided with a decrease in chromatin accessibility for genes associated with pluripotency (e.g., NANOG and RBFOX3). A similar comparison of day 00/day 25 samples showed an uptick in genes known to be expressed in the developing retina (e.g., HES5, SIX3 and BICD1) and a downward trend in the pluripotency markers CER1 and L1TD1 (Figure 5B). A table of the top ten motifs ranked by log10(*p*-value) for both days 00 and 45 were output by HOMER for the investigation of significant motifs. Many of the identified motifs were similar across both timepoints, with some exceptions including Atoh1, NeuroD1 and Neurog2 which were among the top ten enriched motifs at day 45 (Figure 5C). Swarm plots for day 00 and day 45 time points highlight significance (using –log10(*p*-value)) and the number of genes in each group was different between day 00 and 45, particularly in the Homeobox and bHLH TFs motif categories (Figure 5D and Appendix A). To further investigate chromatin changes of key transcription factors/genes, we used IGViewer to overlay ATAC-seq peaks with developmentally relevant TF motifs (Figure 5E). Several genes that participate in retinal development that showed notable peaks at day 45, including ISL1, NEUROG2, and RXRG showed ASCL1 motifs directly within their TSS, suggesting that ASCL1 may play a role in the regulation of these genes. *ASCL1* specifically shows an upregulation in gene expression through day 25, after which its expression decreases to a steady state. Motif enrichment of ASCL1 shows it targets many open transcriptional start sites, suggesting that ASCL1 expression may play a critical role in retinal transcription factor regulatory networks. Overall, this study showed that developmentally relevant motifs were found at different stages of ocular development and this suggests that chromatin accessibility and transcriptional regulation are intimately linked.

## 4. Discussion

Organoid technology offers a unique opportunity to explore human retinal development. In this study, two dual fluorescent reporter hPSC lines were developed to visually track retinal development from the earliest stages of pluripotency to the onset of retinogenesis. The combination of SIX6-GFP and VSX2-tdTomato (or h2b-mRuby3) reporters improved the accuracy and reproducibility of this 3D retina model system by eliminating most non-retinal tissues at early stages. Using this approach, we were able to characterize gene expression and regulatory changes that occurred during the transition of PSCs through early retinal lineage specification. Bioinformatic analysis of developing optic vesicles identified pathways, such as signaling pathways of pluripotency and Wnt signaling, that participate in early retinal differentiation. While other studies have characterized gene expression in retinal organoids, the current study is unique in that it integrates chromatin accessibility with gene expression with an emphasis on early eye development.

Organoid variability within and between experiments complicates efforts to recapitulate human retinal development and disease. In some cases, variability is intrinsic to the cell lines being studied [59,60]. In other instances, it is attributed to differences in the method of retinal differentiation [4,61,62]. Our approach using stage specific reporters allowed us to overcome much of the variability. The benefit of stringent exclusion criteria afforded by reporter expression was that we were able to investigate small incremental changes in gene expression and chromatin accessibility at critical stages of early neural and ocular development. This will enable more precise mechanistic studies that are critical to deciphering human retinal specification and development.

RNA- and ATAC-seq are complementary approaches that enable gene regulatory studies, yet the exact nature of these interactions can be complex. While SIX6-GFP fluorescence, mRNA expression and chromatin accessibility have a high concordance, *VSX2* transcripts and open peak levels at early stages was surprisingly low given the apparent observed tdTomato fluorescence signals. Although VSX2 expression is nuclear, tdTomato is distributed throughout the cell and is quite stable [63], thus *VSX2* levels may appear more abundant than in actuality. Considering that the number of VSX2-tdTomato+ cells in organoids is still relatively low at day 25 (Figure 1C,D) the number of reads could also be masked by other cells in the organoid. This was largely confirmed by the higher expression at later time points. Another study noted a similarly low proportion of VSX2+ cells at the same time point [64]. Regarding the lack of open chromatin peaks, other studies in the mouse retina have also documented a similar disconnect between gene expression and chromatin accessibility; [22] one study even suggested that the chromatin accessibility profile of the retina does not always match the retinal cell-type abundance [20]. Combining RNA-seq and ATAC-seq in FACS sorted or single-cell retinal cells may add clarity to this.

In addition to interpreting functionally accessible chromatin sites, ATAC-seq can also be used to search for TF motifs that bind to specific promoter, enhancer and repressor sequences [65]. Motif enrichment with HOMER highlighted that bHLH and Homeobox TF motifs appeared more frequently at day 45 than day 00 which indicated the relevance of these motifs in retinal development. BHLH TFs participate in many aspects of retinal development including cell fate specification, cell cycle exit and neuronal maturation [66]. Thus, it is not surprising that motifs in this family are more prevalent during development. Since accessibility of chromatin alone does not implicate high gene expression, we correlated highly enriched TF motifs with the open peaks and genes. Only a few genes/TFs (e.g., NEUROG2 and NEUROD1) had significantly positive correlations. Interestingly, TCF4 was strongly negatively correlated, which signals the possibility of TF binding sites being accessible while their corresponding genes have low expression levels due to possible repressor activity at those sites. These findings collectively indicate the importance of utilizing both ATAC and RNA-seq for homing in on functionally relevant chromatin regions. Even still, ATAC- and RNA-seq in parallel leaves some ambiguity, as some target cell populations may be challenging to identify by bulk sequencing. Combining single cell or nucleus RNA sequencing with ATAC data can help reveal gene regulation and cellular heterogeneity. In the future, studies incorporating ChIP-seq can also be performed to further refine and validate the TF motifs that drive human retinogenesis.

SOX2 in pluripotency and neural progenitors has been extensively studied [67,68] where it appears to bind to DNA in a context specific fashion. It only binds to the Nanog promoter in ESCs and to the Egr2 (Krox20) promoter in neural progenitors [69]. Therefore, it is not surprising that our own data confirmed the presence of open chromatin peaks associated with OCT4-SOX2-TCF-NANOG motifs in stem cells but not in retinal organoids. In the developing mouse retina, SOX2 also plays an important role and cooperates with Wnt signaling during establishment of the neurogenic boundary between the ciliary epithelia and neural retina [70]. In addition, deletion of SOX2 in optic cup progenitors leads to a complete loss of retina in favor of an expanded ciliary margin, much like the phenotype resulting from constitutive expression of Wnt-β-Catenin. How it does so in the retina is not completely known, however, possible clues might come from the observation that SOX2 ChIP-seq reads align around *FoxA2*, *Neurog2* and *Axin2* sites [71]. In the hippocampus, SOX2 is known to bind to bivalently marked promoters of ‘poised’ proneural genes such as *Ngn2* and *NeuroD1* at the outset of neural differentiation [72]. SOX2 ChIP-seq in developing retinal organoids could provide new insight into the interplay between *SOX2* and *NEUROG2* in the human visual system. In terms of progenitor competence, numerous retinal progenitor genes showed elevated expression during optic vesicle development including *ASCL1*, *HES1*, *HES5*, *NEUROG2*, *LHX2*, *NOTCH1*, *PAX6*, *RAX*, *SIX3* and *SUFU*. How these genes interact, what cells they are expressed in remains an open question. Future studies incorporating single cell sequencing trajectory analysis might help to resolve these unanswered questions.

Wnt signaling plays crucial roles in tissue specification and differentiation as well as in ocular development [73]. Gradients formed from differential signaling of Wnt family members aid in the maintenance of retinal cell progenitors as well as differentiation [74]. Pathway analysis identified numerous Wnt signaling components that were differentially expressed from stem cells (day 00) to early retinal organoids (day 25) (Figure 3C). *AXIN2*, which plays a role in ocular development [43] and signaling in stem cells [44] was one of the highly enriched genes that was identified as both having a DAR and being a DEG. Its role in ocular development is highlighted by observations that loss-of-function leads to microphthalmia, coloboma, lens defects and expanded ciliary margin. Wnt signaling also plays a role in neurite growth in the retina [75]. “Axon guidance” was also identified as a significant pathway for comparisons between days 00/15 and days 15/25 and although axons do not exist at these early timepoints, many of these genes also participate in neurite outgrowth.

Overall, the combination of RNA-seq and ATAC-seq allowed us to confirm that key developmental mediators of retinal formation correlated with open chromatin regions of the genome. This framework can be expanded in future studies to incorporate longitudinal studies of gene regulation during retinogenesis, with an emphasis on the origins of different cell types.

## Figures and Tables

**Figure 1 cells-11-03412-f001:**
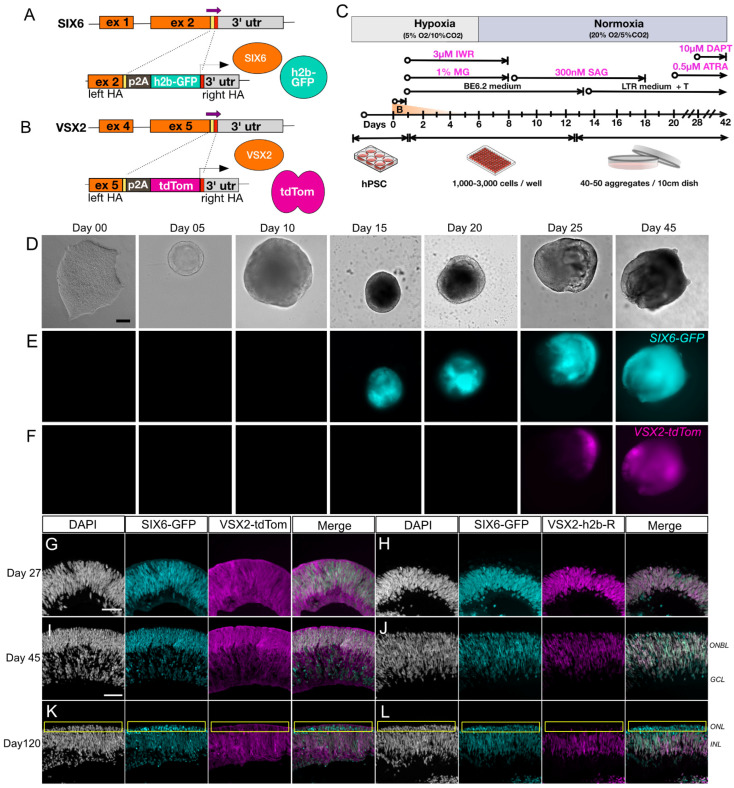
Generation and characterization of a dual fluorescent human PSC reporter line for visualization of retinal differentiation. (**A**) A PSC line with a single *SIX6*-h2b-GFP reporter was (**B**) further modified with a *VSX2*-p2A-tdTomato reporter cassette. (**C**) Graphical protocol for differentiation of retinal organoids adapted from Wahlin et al., 2017. Representative images for differentiating *SIX6*-GFP/*VSX2*-tdTomato PSCs organoids in brightfield (**D**) and (**E**,**F**) *SIX6*-GFP and *VSX2*-tdTomato fluorescence visualized up to differentiation day 45. Organoids at day 15 are smaller than at day 10 due to manual cutting on day 10. Images are separate organoids and are not tracked over time. (**G**–**L**) Cross sections of organoids from day 27 (**G**,**H**), day 45 (**I**,**J**), and day 120 (**K**,**L**) show nuclear GFP expression throughout the organoid and tdTomato expression becoming restricted over time. Panels (**G**,**I**,**K**) represent cytoplasmic *VSX2*-tdTomato reporter expression, while panels (**H**,**J**,**L**) represent nuclear *VSX2*-mRuby3. A yellow box (**K**,**L**) indicates the outer nuclear layer. Scale bars (**D**–**F**) = 100 µm; *VSX2*-tdTom = *VSX2*-tdTomato, *VSX2*-h2b-R = *VSX2*-h2b-mRuby3; Scale bar (**G**–**L**) = 50 µm.

**Figure 2 cells-11-03412-f002:**
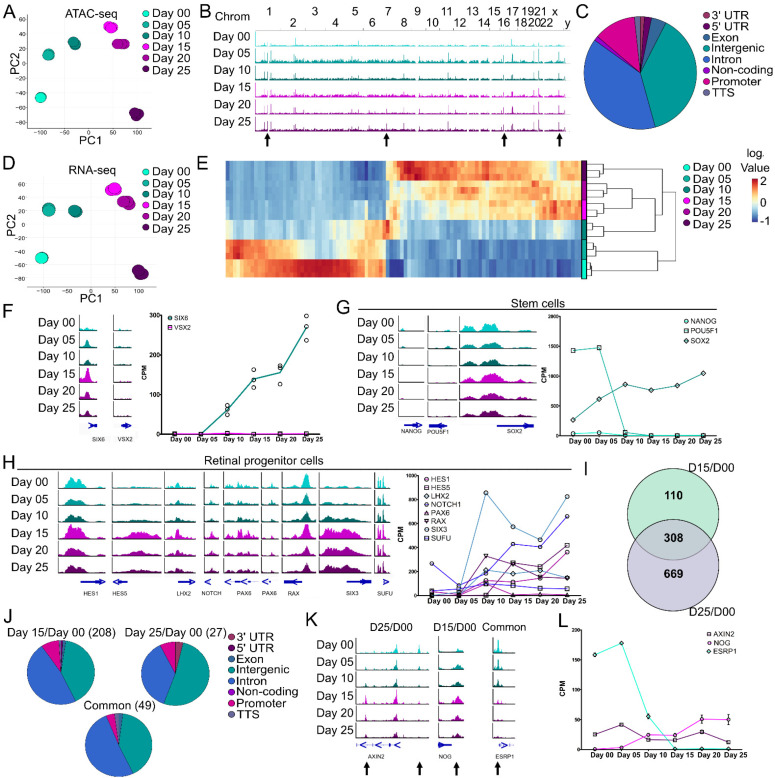
Chromatin accessibility and transcriptomic differences during retinal organoid formation from PSCs. IMR90.4-*SIX6*-GFP/*VSX2*-tdTomato stem cells were differentiated into organoids up to day 25 and pooled (n = 3 replicates; 8–20 organoids/replicate) every 5 days for either ATAC- or RNA-seq analyses. (**A**) Principal component analysis (PCA) of ATAC-seq samples based on 62,201 peaks show a developmental progression over time with little variability between replicates. (**B**) Genomic ATAC peak tracks of the whole genome for days 00 through 25 indicate time-point specific changes (arrows) in chromatin accessibility. (**C**) A pie chart of global genomic annotations based on ATAC-seq analysis from days 00 through 25 identified a high proportion of intergenic, intronic, and promoter peaks. (**D**) PCA of RNA-seq samples from days 00 through 25 based on 12,512 expressed genes show a developmental progression over time with little variability between replicates. (**E**) A heatmap of the top 100 most significant differentially expressed genes between days 00 and 25 (Appendix A). (**F**) Genomic tracks of *SIX6* and *VSX2* at the transcriptional start site (TSS) (left) and expression values from RNA-seq (right) show an increase in *SIX6* TSS chromatin accessibility and gene expression but little/no changes in *VSX2*. (**G**) Genomic tracks (left) and with mRNA expression line graphs (right) of characteristic stem cell markers. (**H**) Genomic tracks (left) and mRNA expression line graphs (right) of retinal progenitor cell markers. (**I**) A Venn diagram depicting DEG/DARs between different days 00, 15, and 25. Numbers indicate the number of differentially expressed/accessible DEG/DARs identified. (**J**) A pie chart of genomic annotations between the different time points from the comparison of ATAC- and RNA-seq. (**K**) Genomic tracks and (**L**) mRNA expression line graphs from DEGs identified (**I**) and their corresponding DARs (arrows). Scale for all genomic tracks = 0–100.

**Figure 3 cells-11-03412-f003:**
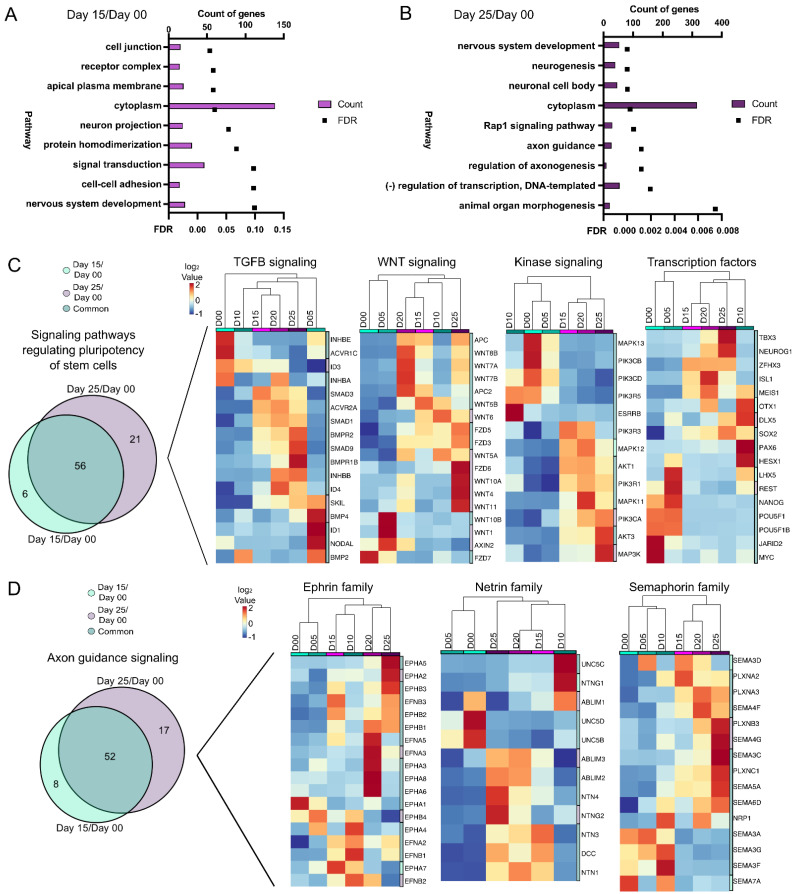
Pathway analysis of gene expression differences from PSCs through early retinal differentiation. (**A**,**B**) Pathway analysis showing the number of DEG/DARs and the FDR for the 9 statistically significant pathways between day 15/day 00 (**A**) and for 9 of the most significant pathways between day 25/day 00 (**B**) (FDR < 0.1). Bar graphs represent the number of DEG/DARs while line graphs represent the FDR. Further analysis was then done using only the DEGs (**C**,**D**). (**C**) “Signaling pathways regulating pluripotency of stem cells” was one common pathway between day 15/day 00 and day 25/day 00 based upon DEG expression. A Venn diagram of the DEGs detected in “Signaling pathways regulating pluripotency of stem cells” at day 25/day 00 and day 15/day 00 time points identified 6 genes unique to day 15/day 00, 21 unique genes in day 25/day 00, and 56 genes common between the comparisons ((**C**), **left**). The DEGs were further classified into hierarchical cluster analyses of days 00 through 25 of TGFβ signaling, Wnt signaling, kinase signaling, and transcription factors and heatmaps of these genes were generated ((**C**), **right**). (**D**) “Axon guidance” was also a pathway in common between day 15/day 00 and day 25/day 00 based on DEGs. A Venn diagram of the DEGs identified in “Axon guidance” at the day 25/day 00 and day 15/day 00 time points identified 8 unique DEGs in day 15/day 00, 17 unique DEGs in day 25/day 00, and 52 DEGs in common between the comparisons ((**D**), **left**). DEGs involved in “Axon guidance” were further categorized into hierarchical cluster analyses of ephrin, netrin, and semaphorin families of genes and heatmaps of these genes were generated ((**D**), **right**).

**Figure 4 cells-11-03412-f004:**
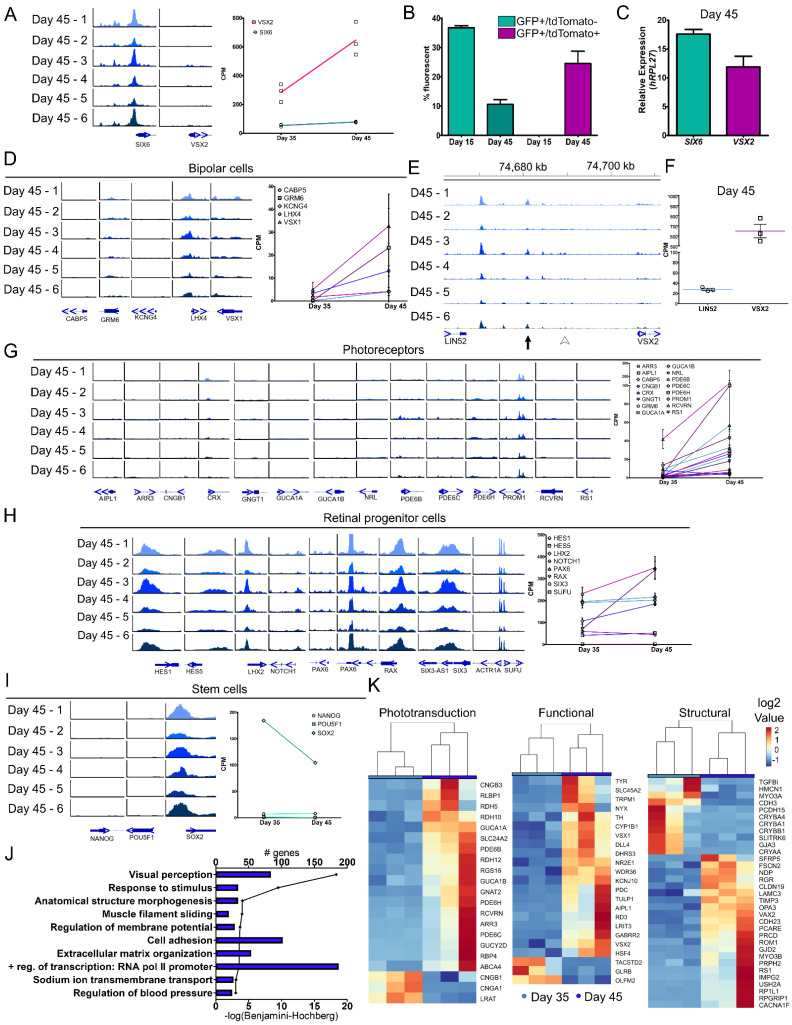
Molecular changes in retinal organoids at the juncture of retinal cell specification. (**A**) Genomic tracks for individual day 45 organoids at the TSS of *SIX6* and *VSX2* (left) and the corresponding RNA-seq values of day 35 and 45 (right). (**B**) Flow cytometry was performed at days 15 and 45 for GFP and tdTomato. No tdTomato+ cells were detected at day 15, and at day 45 all tdTomato+ cells additionally were GFP+. (**C**) qRT-PCR for SIX6 and VSX2 mRNA expression at day 45 was performed. (**D**) Genomic tracks (left) and mRNA expression (right) of bipolar cell genes. (**E**) Genomic region upstream of *VSX2* from day 45 ATAC-seq samples. The location of published core regulatory circuit super enhancer (CRC-SE, Diri et al.) are indicated by black arrows and the location of proximal promoter elements (Kim et al., arrowhead) are indicated by white arrowheads. (**F**) A scatter plot showing corresponding values from RNA-seq of *VSX2* and the upstream gene *LIN52* (right). (**G**) Genomic tracks (left) and mRNA expression line graphs (right) of characteristic photoreceptor markers. (**H**) Genomic tracks (left) and corresponding mRNA expression line graphs (right) for retinal progenitor cell markers. (**I**) Genomic tracks (left) and mRNA expression line graphs (right) from day 45 samples for stem cell markers. (**J**) A bar graph showing the number of genes involved in each pathway and a line graph depicting the –log(Benjamini-Hochberg) for the 10 statistically significant pathways (−log(Benjamini-Hochberg) > 3) as determined by DAVID (Appendix A). (**K**) “Visual perception” was the most highly significant pathway and was further classified into phototransduction, functional, and structural genes and heatmaps were generated. Scale for all genomic tracks = 0–100.

**Figure 5 cells-11-03412-f005:**
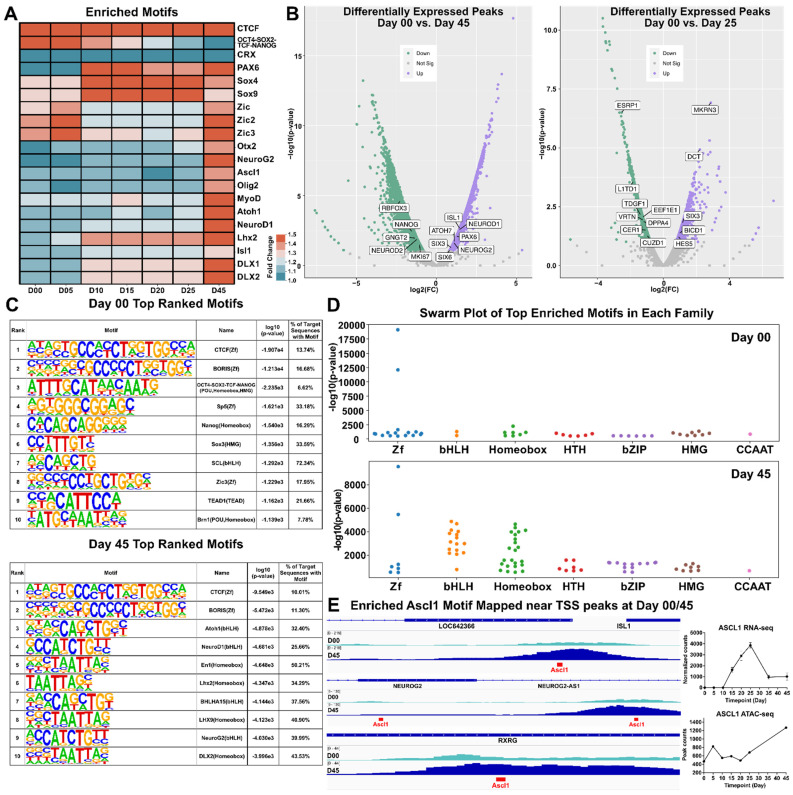
Motif Discovery. (**A**) Heatmap showing the normalized −log10(*p*-values) of selected significantly enriched motifs detected by HOMER. (**B**) Volcano plots displaying showing differentially expressed peaks from Day 45 with respect to Day 00 (left) and Day 25 with respect to Day 00 (right). (**C**) Tables displaying the top 10 motifs each for Day 00 (top) and Day 45 (bottom), along with their motif name and type and the percentage of target sequences that contained the motif. Motifs were ranked according to −log10(*p*-values). (**D**) Swarm plots indicating clusters of top enriched motifs in each transcription factor family plotted according to their −log10(*p*-value) at day 00 (top) and day 45 (bottom). ((**E**), **left**) Genomic tracks showing enrichment of Ascl1 motif (in red) amongst transcriptional start sites (TSS) of *ISL1*, *NEUROG2* and *RXRG* at day 00/45. ((**E**), **right**) RNA-seq normalized counts and ATAC-seq peak counts for *ASCL1*. Zf = zinc finger; bHLH = basic helix-loop-helix; HTH = helix-turn-helix; HMG = High mobility group; bZIP = basic leucine zipper; CCAAT = CCAAT box.

## Data Availability

RNA sequencing datasets are available as raw FASTQ files accessible at the Sequence Read Archive (SRA; #PRJNA771414; PRJNA876944). For reproducibility, detailed Jupyter notebook files containing Python code and R scripts are available on Github (https://github.com/WahlinLab/Retinal-Organoid-ATAC-RNAseq_Cells_2022, accessed on 24 October 2022).

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
