# Peer review of "Chromatin Accessibility and Transcriptional Differences in Human Stem Cell-Derived Early-Stage Retinal Organoids"

_cells, 2022, doi:10.3390/cells11213412_

Round 1
Reviewer 1 Report
Major :
1- Line 108 : precise primers used to amplifying the 1,157 bp fragment of the VSX2 locus from genomic DNA. Please specify human genomic DNA.
2- Figure 1A: Six6-GFP is a fusion protein? If yes please link SIX6 and GFP in the picture
3- Figure 1B: Vsx2 have 5 exons. Please explain how you integrate tdTom cassette after the exon2 and a hypothetic 3’UTR. Or you want to say Exon 4 and Exon 5 in your schema? VSX2-GFP is a fusion protein? If yes please link VSX2 and GFP in the picture.
Minor:
- Line 100 : These were, were maintained
- Text/figure: Please harmonize VXS2 cyto, p2A or td tomato, tdtom between the text and the figure. Same think for VSX2 H2B, ruby…
Author Response
Reviewer 1:
Major:
Q1) Line 108 : precise primers used to amplifying the 1,157 bp fragment of the VSX2 locus from genomic DNA. Please specify human genomic DNA.
A1) Thank you for identifying this missing information. We have added the primer sequences for amplifying the VSX2 locus, and clarified that this is from human genomic DNA (line 109).
Q2) Figure 1A: Six6-GFP is a fusion protein? If yes please link SIX6 and GFP in the picture
A2) Thank you for your thoughtful review of the diagram. The SIX6-GFP reporter line was designed as a p2A-h2b-eGFP from our previous publication (Wahlin et al., 2021, reference 10), and the diagram in Figure 1A is correct in showing that SIX6 and GFP are separated. To clarify this we have modified the figure to include a p2A sequence. We have also added the reference of how the SIX6-GFP reporter line was designed (reference 10, line 121), and hope this helps clarify any confusion.
Q3) Figure 1B: Vsx2 have 5 exons. Please explain how you integrate tdTom cassette after the exon2 and a hypothetic 3’UTR. Or you want to say Exon 4 and Exon 5 in your schema? VSX2-GFP is a fusion protein? If yes please link VSX2 and GFP in the picture.
A3) Thank you again for reviewing the diagrams and apologies for the mistake. This should be exons 4 and 5 in the diagram, and Figure 1B has been edited for accuracy. Similar to the SIX6-GFP reporter line, the VSX2-tdTomato reporter was generated via a p2A (ribosomal skipping) sequence and not as a fusion protein so the diagram is correct in showing the separation of VSX2 and tdTomato fluorescence.
Minor:
Q1) Line 100 : These were, were maintained
A1) Thank you finding this typo. We have edited the text on line 100 to “These were maintained…”
Q2) Text/figure: Please harmonize VXS2 cyto, p2A or td tomato, tdtom between the text and the figure. Same think for VSX2 H2B, ruby…
A2) Thank you for the suggestion. Nomenclature is now harmonized between figures and text.
Reviewer 2 Report
Jones and co-authors described genome-scale analyses of DNA accessibility and transcriptional changes associated with the differentiation of human pluripotent stem cells into retinal cells. The work provides relevant datasets obtained from well-conducted experiments. Here is a list of points to improve the text.
Main points
- The results on SOX2 were interesting and could contribute to the discussion. Would enrichement for SOX2+ cells contribute to a more pure population of retinal cells? Earlier or later in the organoid protocol?
- What would be the advantages /disadvantages of using a CRX reporter gene?
- The discussion focused more on the methods rather than the biology. What were other biological insights provided by this work besides Wnt signaling? Authors may include more perspectives on follow-up studies.
Minor points
Line 28: "at day 25, respectively";
Line 76: "retinal cells' or "retinal fate" seems more appropriate than "bonafide retina";
Line 97: "Culture of Pluripotent stem cells"
Lines 408-409: this statement of "stemness genes" seems misleading by suggesting similar roles in pluripotent and neuronal-like cells. Perhaps Sox2 expression relates to a neural progenitor-like (or adult stem-like) state, as stated earlier in the text.
Line 477: lacks the reference
Line 489: typo "*";
Line 571: ASCL1 instead of Ascl1;
Line 589: "Organoid technology offers..."
Author Response
Reviewer 2:
Main points:
Q1) The results on SOX2 were interesting and could contribute to the discussion. Would enrichement for SOX2+ cells contribute to a more pure population of retinal cells? Earlier or later in the organoid protocol?
A1) We agree that it is an interesting topic and as such we have added a section on SOX2 in the methods section that provides links known developmental regulators, including NEUROG2. Regarding the question about cell purity, deletion would clearly be deleterious since it has already been shown to result in small eye phenotypes, however, we would not expect that over expression would increase the purity of retinal cells since SOX2 is already broadly expressed in neural progenitors, including the retina.
Q2) What would be the advantages /disadvantages of using a CRX reporter gene?
A2) Since CRX is expressed early in photoreceptor development it would allow for detection of early changes in chromatin accessibility and gene expression that could provide useful clues about cone and rod specification if coupled to single-cell trajectory analysis. This is now reflected in the text. However, due to the timeline of this study focusing on early retinal development, we chose to not focus on CRX expression as photoreceptors do not mature until later on at a timepoint outside the scope of our study.
Q3) The discussion focused more on the methods rather than the biology. What were other biological insights provided by this work besides Wnt signaling? Authors may include more perspectives on follow-up studies.
A3) Combined with Q1 we have added some discussion of SOX2 and NEUROG2 which are important regulators of both neural progenitors and early retinal development.
Minor points
Q1) Line 28: "at day 25, respectively";
A1) Thank you for your comment. We agree with your statement and edited line 28 to read “…day 15 and tdTomato (or mRuby3) at day 25, respectively.”
Q2) Line 76: "retinal cells' or "retinal fate" seems more appropriate than "bonafide retina";
A2) Thank you for clarifying this statement. We have edited line 76 to read “VSX2-tdTomato; retinal lineage”. We hope this clarifies the use of VSX2 in retinal differentiation.
Q3) Line 97: "Culture of Pluripotent stem cells"
A3) Thank you for recommending changing the title. We have accepted the edit and line 97 now reads “Culture of pluripotent stem cells”
Q4) Lines 408-409: this statement of "stemness genes" seems misleading by suggesting similar roles in pluripotent and neuronal-like cells. Perhaps Sox2 expression relates to a neural progenitor-like (or adult stem-like) state, as stated earlier in the text.
A4) Thank you for pointing this out. We agree with you that this is misleading and have changed it to say a “progenitor-like state” in order to remove ambiguity.
Q5) Line 477: lacks the reference
A5) Thank you for identifying this missing information. We have added the Kim et al reference to line 479.
Q6) Line 489: typo "*";
A6) We appreciate your thorough examination of the text. In Figures S3E and S3G, we use the “*” symbol for referencing the Diri et al. published location of the VSX2 peaks; however, the figure legend states that we are using arrowheads for the peaks. We have reconciled the figures and legends to correspond to the Diri et al. peaks as “*”. We hope this clarifies any confusion.
Q7) Line 571: ASCL1 instead of Ascl1;
A7) Thank you for finding this error. We have edited line 573 to read “…RXRG showed ASCL1 motifs…”
Q8) Line 589: "Organoid technology offers..."
A8) Thank you for recommending this new text. The authors agree with the edit, and have made the change to be “Organoid technology offers…”
Round 2
Reviewer 1 Report
ok.